# Chaperonology: The Third Eye on Brain Gliomas

**DOI:** 10.3390/brainsci8060110

**Published:** 2018-06-14

**Authors:** Francesca Graziano, C. Caruso Bavisotto, A. Marino Gammazza, Francesca Rappa, Everly Conway de Macario, Albert J. L. Macario, Francesco Cappello, Claudia Campanella, Rosario Maugeri, Domenico Gerardo Iacopino

**Affiliations:** 1Department of Experimental Biomedicine and Clinical Neuroscience, Section of Neurosurgery, University of Palermo, 90127 Palermo, Italy; rosario.maugeri1977@gmail.com (R.M.); gerardo.iacopino@gmail.com (D.G.I.); 2Department of Experimental Biomedicine and Clinical Neuroscience, Section of Human Anatomy, University of Palermo, 90127 Palermo, Italy; celestebavisotto@gmail.com (C.C.B.); antonella.marino@hotmail.it (A.M.G.); francyrappa@hotmail.com (F.R.); francapp@hotmail.com (F.C.); claudiettacam@hotmail.com (C.C.); 3Euro-Mediterranean Institute of Science and Technology (IEMEST), 90136 Palermo, Italy; ajlmacario@som.umaryland.edu; 4Institute of Biophysics, National Research Council, 90143 Palermo, Italy; 5Department of Microbiology and Immunology, School of Medicine, University of Maryland at Baltimore-Institute of Marine and Environmental Technology (IMET), Baltimore, MD 21202, USA; econwaydemacario@som.umaryland.edu

**Keywords:** high-grade gliomas, molecular chaperones, heat shock proteins, neuroimaging, neuromonitoring, chaperonology, chaperonotherapy

## Abstract

The European Organization for Research and Treatment of Cancer/National Cancer Institute of Canada Phase III trial has validated as a current regimen for high-grade gliomas (HGG) a maximal safe surgical resection followed by radiotherapy with concurrent temozolamide. However, it is essential to balance maximal tumor resection with preservation of the patient’s neurological functions. Important developments in the fields of pre-operative and intra-operative neuro-imaging and neuro-monitoring have ameliorated the survival rate and the quality of life for patients affected by HGG. Moreover, even though the natural history remains extremely poor, advancement in the molecular and genetic fields have opened up new potential frontiers in the management of this devastating brain disease. In this review, we aim to present a comprehensive account of the main current pre-operative, intra-operative and molecular approaches to HGG with particular attention to specific chaperones, also called heat shock proteins (Hsps), which represent potential novel biomarkers to detect and follow up HGG, and could also be therapeutic agents.

## 1. Introduction

Gliomas and other neuroepithelial tumors make up 49% of primary brain tumors, and meningiomas are the next most frequent histologic type (27%) [1,2].

Glioma tumor cells display histological similarities to normal glial cells, including astrocytes and oligodendrocytes. Consequently, they are classified as astrocytoma, oligodendroglioma, or oligoastrocytoma. The 2007 World Health Organization (WHO) classification categorized gliomas as low-grade (WHO grade I and II) and high-grade (WHO grade III and IV). More than half of all gliomas are GBM (glioblastomas multiforme) (WHO grade IV astrocytoma).

Genetic studies on the development of brain tumors have identified a number of recurrent chromosomal abnormalities and genetic alterations, particularly in malignant gliomas such as GBM. The gold standard treatment, currently in use, is optimal safe surgical resection followed by adjuvant partial brain radiotherapy with concurrent temozolomide, and the subsequent continuation of temozolomide for six cycles. The natural history remains extremely poor; indeed, the overall survival is usually only around 12 months and the overall 5-year survival rate is less than 5% [1,2].

An important prognostic factor in oncological neurosurgery is the extent of resection (EOR) [3,4,5,6,7]. Tumor visualization is the crucial factor to maximize the EOR and it is accomplished by the employment of different tools such as: neuronavigation, fluorescence, and intra-operative imaging, including magnetic resonance imaging (MRI), computerized tomography (CT), and ultrasound (US) [8,9,10].

Quality of life may be affected by potential post-operative neurological complications, which could also defer the initiation of adjuvant therapy, worsening the survival rate. Tumors involving eloquent brain areas have been considered “high risk” for resection in terms of potential risk for neurological morbidity. Multiple pre-operative techniques are nowadays in use to help identify eloquent areas and their relationships to brain lesions, such as functional magnetic resonance imaging (MRI), diffusion tensor imaging (DTI), transcranial magnetic stimulation (TMS), magnetoencephalography, and magnetic source imaging (MSI). Specific intra-operative tools, such as direct electrical stimulation (DES) mapping, intra-operative MRI or computerized tomography (CT), and 5-aminolevulinic acid (5-ALA), may be also employed to maximize the tumor resection while assuring the preservation and safety of the eloquent areas [11,12,13,14].

The dismal clinical outcome of gliomas has made high-grade gliomas (HGG) an urgent subject of cancer research for the identification of novel factors associated with glioma development.

Among the various factors that participate in brain carcinogenesis, molecular chaperones, also known as heat shock proteins (Hsps) are, nowadays, the focus of attention because they are believed to play crucial roles in tumor initiation and progression. Consequently, they are viewed as powerful candidates for biomarkers and as therapeutic targets or agents. Molecular chaperones participate in many physiological cellular networks and in intercellular communication to maintain homeostasis, and to assist other proteins to achieve and maintain a functional conformation, thus regulating cell survival and differentiation. Furthermore, if Hsps are abnormal or malfunctioning, they can contribute to the development of diseases, named chaperonopathies [15]. In view of the increasing importance attributed to Hsps, they have been, and are currently, extensively studied in numerous pathologies, including cancer. In this regard, Hsps have already established themselves as very promising biomarkers of various cancers with applications in diagnosis, assessment of prognosis, and response to treatment.

Hsps are evolutionarily conserved proteins involved in various cellular processes including brain tumors, and variation in their expression seems to be tightly associated with the progressive staging and prognosis of gliomas. It has been demonstrated in several human cancers that Hsps promote tumor growth by stimulating cell proliferation and inhibiting death pathways and it is assumed that in glioma Hsp27 (as well as other Hsps) could induce radioresistance [16]. Increase in the knowledge on the role of Hsps in brain tumors will provide an opportunity to use these molecules as biomarkers in diagnosis, as well as in the assessment of prognosis and response to treatment [15,16,17]. This review aims to introduce Hsps into the field of HGG, to stimulate investigations of their properties and functions in the brain, and to explore their pathophysiological roles. The possibility is open to add molecular determinations of chaperones to the other routine analyses in the assessment of patients during pre-operative, intra-operative, and post-operative evaluation.

## 2. Materials and Methods

Our research of the public databases, mainly PubMed, was initiated on 1 January 2018 with the aim of identifying all studies related to pre-operative neuro-radiological evaluation, to the intra-operative and post-operative molecular assessment in patients affected by HGG.

We performed the literature review on the databases using the following combinations of terms: “aminolevulinic acid” AND “high-grade glioma”, “MRI brain” AND “high-grade glioma”, “fMRI” AND “high-grade glioma”, “DTI” AND “high-grade glioma”, “Neuronavigation” AND “high-grade glioma”, “brain mapping” AND “high-grade glioma”, “CEUS” AND “high-grade glioma”, “Intra-operative CT” AND “high-grade glioma”, “high-grade glioma” AND “neuroimaging”, “HSP” AND “high-grade glioma”. We selected only articles written in English. Retrospective and prospective studies and clinical trials were included, while editorials, case reports, and commentaries were excluded.

The results of the literature review were categorized according to the “curative EYE” to a patient affected by HGG: 1. Pre-operative assessment (FIRST EYE); 2. Intra-operative study (SECOND EYE); 3. Molecular assessment (THIRD EYE).

## 3. Results

### 3.1. FIRST EYE

#### 3.1.1. Pre-Operative Assessment

##### Neuro-Radiological Evaluation MRI Brain, Functional MRI (fMRI), DTI, and Diffusion Tensor Tractography (DTT)

The gold standard in detecting a brain glioma is magnetic resonance imaging (MRI). Specific sequences include a volumetric T1-weighted, Gd-enhanced sequences, FLAIR sequences, and T2-weighted sequences [18]. Functional MRI (fMRI) has become a largely available clinical tool for the pre-surgical evaluation of functional areas prior to brain tumor surgery. It is a non-invasive brain-mapping method to guide neurosurgical treatment decisions. fMRI pinpoints functional networks involved in a determined function such as a motor or language tasks. However, especially near the tumor mass, vascular changes can lead to a neurovascular uncoupling instead of the regular coupling and this could produce false-negative fMRI results, making it unreliable for planning [19].

fMRI is able to localize along the cortical surface each neurological function; however, it is not able to delineate subcortical white matter tracts connecting important cortical areas. 

In the last 10 years, pre-operative functional MRI and DTT became part of the clinical routine to decrease the surgical risk of tumors in eloquent brain areas such as motor, language, and visual cortex areas. In order to reduce the risk to damage the motor, or sensitive, or cognitive pathways, diffusion tensor imaging (DTI) tractography (DTT) is one of the most successful pre-operative examination methods [20]. DTI is an MRI technique that measures water diffusion tensor in living tissues. DTI is sensitive to the diffusion of water molecules. In white matter, the principal direction of this diffusion corresponds with the main fiber orientation within a given voxel. DTT is a method based on diffusion tensor magnetic resonance imaging. Tractography using this method is capable of depicting subcortical white matter tracts in vivo, which is not possible by conventional imaging. DTT may use two different algorithms: the deterministic and the probabilistic methods [20,21]. The probabilistic method is able to determine the probabilistic connectivity of different brain areas; thus, it can identify the subcortical nuclei based on their cortical connections. Using this technology, it is possible to determine the individual anatomy and identify the dislocation of thalamic nuclei, in order to plan the surgical route to the target. Conventional MRI, specifically that depicting the tumor location, is insufficient. By contrast, the pre-surgical DTT of the corticospinal tract and the inclusion in the neuronavigation system are tools extremely useful in promoting a safer and more effective surgical resection and improving the overall functional status.

### 3.2. SECOND EYE

#### 3.2.1. Intra-Operative Assessment

##### 5-A.LA

Fluorescence-guided surgery (FGS) has revolutionized the neurosurgical treatment of brain tumors over the last 10 years. The use of 5-ALA (5-aminolevulinic acid) and FGS in patients with gliomas was described in 1998 [22]. 5-ALA allows intra-operative visualization of the tumor bulk in addition to the surrounding zone of tumor infiltration present in malignant gliomas. 5-ALA-induced fluorescence assists the neurosurgeon, during tumor resection, with real-time information to distinguish tumor from normal tissue independently by neuronavigation and brain shift. 5-ALA is a precursor of the Heme synthesis pathway, which favors the production of protoporphyrin IX (PpIX). This is a molecule that emits fluorescence when excited by an appropriate wavelength; specifically, under the light beam in blue-violet, PpIX emits light in the red region of the visible spectrum, allowing the localization of tumor tissue that would otherwise be difficult to distinguish from adjacent normal brain tissue. Although by this approach it is possible to identify positive 5-ALA areas, it is more accurate on the surface of the tumor mass than in profundity and does not allow the detection of the diseased tissue which is below the tumor mass [22,23].

##### Neuronavigation

Neuronavigation allows the inclusion in the system of acquired images such as those provided by CT, MRI, functional MRI, and DTI to achieve orientation in the surgical field. Neuronavigation is a very helpful tool to guide the surgeon from the planning step of the skin incision through to the microscopic step of tumor removal. However, it is affected by brain shift and brain deformation which progressively make the information provided by neuronavigation worthless [24].

##### Brain Mapping

The concept of the brain connectomics has revolutionized glioma surgery in eloquent hemispheres. Thus, only the functional mapping could be considered the method of choice for determining full tumor removal [25,26]. Indeed, if direct electrical stimulation (DES) demonstrates no functional localization within the tumor, or within portions of the tumor, then resection is performed within the context of maximal safe resection. Thus, lesions viewed by some physicians as “inoperable” or “unresectable” based on imaging studies may very well resected with the use of DES [25,26]. 

Intra-operative localization of eloquent cortex may be achieved through cortical electrical stimulation in awake patients and the somatosensory evoked potential (SEP) phase-reversal technique in sedated patients. In an awake patient, the discrete cortical electrical stimulation remains the gold standard because it can be used to localize a variety of eloquent cortical areas (sensory, motor, and language areas) [25]. In sedated patients, the SEP phase-reversal technique is mainly useful for localizing the motor sulcus, usually around the upper limb somatosensory focus. While other non-invasive, pre-surgical modalities such as functional MRI, TMS, magnetoencephalography, and diffusion tractography may become increasingly useful adjuncts for localizing eloquent areas and pre-operatively assessing surgical risk, intra-operative DES is currently the most accurate and the robust method available for identifying functional brain tissue. While tractography is helpful for defining white matter pathways and guiding the use of subcortical DES, resection limits should be ultimately guided by direct cortical stimulation [27].

##### Intra-Operative CT, MRI

Intra-operative MRI and CT (iMRI and iCT) overcome brain shift and brain deformation and offer high spatial resolution and a wide field of view, but they are expensive and time/space-consuming [28]. They are usually employed directly in theatre, soon after the tumor removal and before the closure step, in order to provide an immediate radiological imaging regarding or residual mass hidden by brain collapse, or recent hematoma formation. Even though they could be useful, they cannot be considered real-time intra-operative imaging modalities since is not possible to operate directly under their imaging guidance.

##### Contrast-Enhanced Ultrasound (CEUS)

Intra-operative ultrasound (iUS) has been used in neurosurgery since the early 1980s, and over the years many applications of this method have been reported. iUS is truly a real-time, dynamic technique that offers a good temporal and spatial resolution [29]. Its high spatial resolution allows accurate tissue differentiation, which has been shown to improve the EOR in glioma surgery. Contrast-enhanced US (CEUS) is an iUS modality that uses an ultrasound contrast agent (UCA) to improve the contrast between tumor, healthy tissue, and artefacts.

CEUS can highlight all glial tumors, particularly GBMs, with a specific contrast enhancement, which also allows their characterization and visualization in the surgical volume. It can overcome the limitations of neuronavigation and may highlight fluorescent tumor areas hidden by brain collapse [29,30,31].

### 3.3. THIRD EYE

#### Molecular Assessment

Although the clinical approach has become ever more definite in the management of patients with high-grade gliomas, molecular profiling has gained acceptance since it enhances the understanding of brain tumor oncogenesis. Advances in the identification and characterization of molecular factors underpinning GBM development will certainly influence progress in designing prognostic and predictive tools and procedures for assessing and predicting clinical outcome. Nowadays, the molecular markers that currently are the most informative include the 1p/19q co-deletion status, Figure 1, which is associated with a best or poor prognosis of patients treated with radiation therapy with or without chemotherapy, whether is co-deleted or not, respectively [32]. The isocitrate dehydrogenase 1/2 (IDH1/2) gene mutation is identified in >70% of WHO grades II and III gliomas and secondary glioblastomas and constitutes a discriminant between primary and secondary GBM [33]. IDH mutations, Figure 1, are associated with a significantly longer survival time compared with IDH wild-type tumors in patients age ≥60 years with anaplastic astrocytoma and glioblastoma; therefore, the absence of this mutation correlates with a poor prognosis [34].

Studies of epigenetic signatures showed that the hypermethylation of CpG islands, Figure 1, is associated with the transcriptional silencing of the gene, therefore the effect depends on the affected gene. In human GBMs, the hypermethylation status varies with glioma subtypes; particularly, secondary GBMs have a higher frequency of promoter methylation than primary GBMs. For instance, the O-6 methylguanine-DNA-methyltransferase (MGMT) promoter methylation, Figure 1, was found in a large percentage of GBM patients and the gene encodes an enzyme which removes alkyl groups from the O-6 position of guanine [35]. Consequently, GBM patients with MGMT hypermethylation showed sensitivity to alkylating agents such as temozolomide, with an accompanying improved outcome [35]. Further genetic alterations in ATRX, TP53, PTEN, EGFR, RB1 NF1, ERBB2, PIK3R1, and PIK3CA are now taken into consideration to guide glioma classification and diagnosis, as well as to program individualized treatments for the distinct molecular subtypes, Figure 1.

In a variety of human cancers, Hsp levels are associated with the prognosis outlook and therapeutic resistance; these are proteins known to promote tumor growth by stimulating cell proliferation and inhibiting cell death pathways [15]. The levels of many Hsps are elevated in various types of cancer and Hsps and their overexpression often indicates a poor prognosis in terms of survival and response to therapies [36].

In glioma pathogenesis, molecular chaperones represent a novel and important research field because they are involved with various roles. Hsp47 and Hsp90 promote angiogenesis in glioma cells lines, while Hsp27, Hsp40, and Hsp70 affect the survival pathway, promoting cancer cell survival [37,38,39,40]. Conversely, the role of Hsp60 in HGG has not yet been fully elucidated. In another cancer type, it has been demonstrated that Hsp60 is upregulated and displays an anti-apoptotic role promoting cell survival [41,42].

Hsp60 has been found to be differentially expressed in glioblastoma cell lines and its functional significance seems to be dependent on its localization [43,44]. Hsp60 is downregulated in glioblastoma multiforme compared with non-tumor tissues [44]. However, to the best of our knowledge, no other study has evaluated Hsp60 expression in brain tumors and our current research aims to clarify its role in brain tumor cells and its microenvironment (Figure 2).

Numerous studies have shown that Hsp60, Hsp70, and Hsp90 are secreted by cancerous cells via exosomes, and can have opposing effects—immunosuppressing or immunostimulating [17,45,46]. Our research group found that exosomal Hsp60 levels in the plasma of patients before colon cancer surgery were significantly higher than in the exosomes from the same patients after tumor ablation [45]. Hsp60 exportation by exosomes suggests the involvement of this chaperonin in inflammation, immune system modulation, and the regulation of the tumor microenvironment and growth [46,47]. Therefore, exosomal Hsp60 may contribute to the regulation of gene expression in target cells at distant sites [48]. In light of the available data on the multiple roles in various organs and tissues, intra- and extracellularly, Hsp60, free or in exosomes, has great potential in glioma management: it could serve as a biomarker to help in differential diagnosis and patient classification, assess tumor grade, and evaluate prognosis and response to therapy. The situation with Hsp60 is particularly promising because it can be measured in blood, as a “liquid biopsy” with minimal discomfort for the patients. Thus, free and exosomal HSP60 can be considered as practically convenient biomarkers that can easily be sampled and analyzed to obtain information on a tumor with a minimally invasive procedure that is still very helpful for clinicians (Figure 1).

## 4. Discussion

Nowadays when we are dealing with a brain glioma, the neurosurgical approach includes a pre-operative clinical and neuro-radiological full assessment with gadolinium, in addition to functional neuroimaging. The mainstay of treatment for newly diagnosed GBM is resection followed by radiation therapy and chemotherapy. A crucial prognostic factor in oncological neurosurgery is the extent of resection (EOR) [6,7,8,9,10]. Neuronavigation is extremely helpful in finding the tumor and the surrounding neurovascular structures, but it is affected by brain shift and brain deformation so it loses reliability during surgery [10,11,13].

In the last 10 years, the advent of fluorescence-guided surgery (FGS) has revolutionized the neurosurgical treatment of brain tumors. During tumor resection, 5-ALA-induced fluorescence supports the neurosurgeon with real-time information for differentiating tumor from normal tissue that is independent of neuronavigation and brain shift. 

As demonstrated by a randomized, controlled phase-III study, 5-ALA administration and FGS provide a more complete resection of malignant gliomas and better progression-free survival. 5-ALA (Gliolan) is a safe compound with only minimal side effects, approved for human use in Europe, Asia, and Australia. It has been used in thousands of patients worldwide (information provided by Medac, Wedel, Germany) [22,23].

Intra-operative US truly is a real-time, dynamic technique that offers a good temporal and spatial resolution. Its high spatial resolution permits an accurate tissue differentiation, which has been shown to improve the EOR in glioma surgery. Contrast-enhanced US (CEUS) is an iUS modality that uses an ultrasound contrast agent (UCA) to improve the contrast between tumor, healthy tissue, and artefacts. It can overcome the limitations of neuronavigation since it truly is a dynamic technique and may highlight fluorescent tumor areas hidden by brain collapse, following tumor removal [11].

In cases of tumor involvement of functional brain areas, intra-operative brain mapping is employed to allow for a safe tumor removal. It improves the extent of resection while decreasing the risk of post-operative deficits, even for tumors located in or close to functional areas [25,26,27].

The conventional treatment strategy for glioma mainly entails maximal surgical abscission, radiotherapy, and concomitant/adjuvant chemotherapy [5,6,7,8]. Despite the improvement of the current approach for a patient’s treatment, the prognosis for GBM patients remains poor because of tumor genetic and phenotypic heterogeneity, multiple activation of key oncogenic pathways, acquired therapeutic resistance, and cytoprotective mechanisms in GBM cells. 

Concerning the molecular management of tumors, the approach that takes into account the multifaceted role of molecular chaperones is increasingly acknowledged. In the field of biomedical research, “chaperonology” studies molecular chaperones and the possible malfunctioning of them, which give rise to a variety of pathological conditions known as chaperonopathies [15]. Chaperone therapy, chaperonotherapy, involves the use of chaperones in the treatment of chaperonopathies [15]. 

Our research group has studied the chaperone Hsp60 in detail and we have demonstrated the overexpression of this protein during human carcinogenesis [44,45,47]. Hsp60 can interact directly with molecules in various cell compartments and can also be found on the membrane surface of normal and tumor cells. Hsp60 regulates proteins involved in apoptosis and cell cycle and when it is dysregulated it can promote disease, such as cancer. The role of Hsp60 in carcinogenesis depends on the tumor type and must be determined accordingly, namely within the context of the tumor under consideration. Our recent research, as well as the research of other groups, has been directed to strongly demonstrate that Hsp60 is released by both normal and pathological cells, but the real mechanisms by which this protein is translocated outside the cell are not yet completely clear. Our study proposes that Hsp60 release into the extracellular space is the result of an active secretion mechanism—not a passive phenomenon due, for example, to cell damage or death with membrane disruption, but rather a reflection of a general physiological event [48]. Based on the results of our in vitro studies, our group proposed a multiphase process to explain the translocation of Hsp60 from the cytosol to the extracellular space that includes a combination of protein traffic pathways (reviewed in Reference [48]). Hsp60 in normal cells localizes mainly in mitochondria, while in various tumor cells it also accumulates in the cytoplasm and reaches the cell membrane and the Golgi. At the membrane, lipid rafts internalize Hsp60 into multivesicular bodies (MVB) which blend with the plasma membrane, releasing the content via exosomes. In these, it is located in the membrane and probably also inside. Hsp60-loaded exosomes thus would reach other cells near and far through the circulation [44,45,47]. 

Chaperones are also activated during normal cellular physiology. They assist other proteins in their folding and re-folding and, when the proteins are defective or misfolded beyond repair, chaperones direct them to degradation. They mediate protein trafficking inside the cell, avoiding irregular aggregations and mismatched proteins interactions. Some chaperones have anti-apoptotic properties and have been found to be elevated intracellularly in many human cancers. They are also secreted outside the cell. They have also been found to be elevated extracellularly, e.g., blood, in a variety of human cancers. They are represented inside cells as well as outside, circulating in blood free or in extracellular vesicles, such as exosomes [47,48]. Chaperones localized on the surface or in the inner part of exosomes, secreted by normal and tumor cells, could be key players in intercellular communication. Exosomal chaperones offer significant chances for clinical applications, including their use as biomarkers for diagnostic and monitoring purposes and for therapeutic applications and drug delivery. Although some cancer-associated miRNAs related with chaperone expression and regulation in other tumor types, such as breast cancer, have been already identified, the same cannot be said for GBM. In our opinion, this ambitious characterization would aid in the efficient design of new anti-glioma therapeutics.

When patients are treated with conventional therapy, as chemo/radiotherapy, they often develop chemo/radioresistance. Furthermore, an increase in Hsps levels has been observed that correlates with the expression of epithelial to mesenchymal transition (EMT) markers. This suggests that Hsps play a role in cancer resistance, for instance as an anti-apoptotic factor, promoting the cancer cells’ survival [46]. Working in this manner, it could be assumed that the probable key factors in the failure of therapies against GBM are Hsps, and the novel challenge for therapeutic interventions in glioma management is without a doubt the molecular approach based on the characterization of Hsps and their regulation, influencing the cellular transformation and cancer progression.

Nowadays when we are dealing with brain glioma, the neurosurgical approach includes clinical and radiological assessment, followed by a surgical treatment which would consider a optimal total resection using 5-ALA, neuronavigation, and neurophysiological brain monitoring in order to assure a safe, total, and satisfying treatment. In the post-operative period, chemotherapy and radiotherapy should be considered and in the follow-up stage the patient should be monitored both clinically and radiologically.

Considering gliomagenesis, it would be advisable to look at the brain tumors through the Chaperone Eye, in view of think about Hsps as biomarkers for diagnosis, prognosis assessment, and follow-up, as well as promising therapeutics targets [17,44,45,46,47,48,49] in future clinical applications.

The interest of the scientific community in the molecular tumoral field will induce us in the future to include molecular analyses in pre-, intra- and post-operative evaluations, including the quantification and characterization of circulating Hsp60 free and in exosomes, as well as in biopsy specimens when available. 

## 5. Limitations of the Study

Few studies have evaluated the relationship between gliomas and Hsps. Hsp60 is upregulated in some human cancers, including some cases of glioblastoma, and orchestrates a cytoprotective pathway that involves the stabilization of survivin levels and the restraint of p53 function [50]. High expression of Hsp60 was detected in nearly all tumors studied, both in high-grade gliomas and meningiomas [51]. Also, Hsp40 (DnaJ), Hsp70, and Hsp90 were found to be elevated in all brain tumors [51]. There is evidence of a positive correlation between Hsp levels and brain tumor progression, which points to the distinct possibility of using Hsps as biomarkers or as components of antitumor vaccines [51,52]. Further studies involving large number of patients are needed to clearly define the relationship between Hsps and tumor aggressiveness and prognosis. It is possible that some Hsps are more specific for a tumor type while others might be so for other types. It follows that the elucidation of Hsp-tumor specificity is a promising research line to standardize study protocols focusing on specific cases (personalized medicine).

## 6. Conclusions and Future Perspectives

There currently is sufficient information to consider molecular chaperones—Hsps, e.g., Hsp60—as promising biomarkers for the early diagnosis and follow-up of brain tumors as well as for potential therapeutic targets in those cases in which the chaperone favors carcinogenesis and, therefore, the chaperone has to be blocked or eliminated (negative chaperonotherapy). Contrariwise, in those cases in which the chaperone fails to stop carcinogenesis due to deficient function, positive chaperonotherapy would be indicated, namely the defective chaperone should be boosted to restore its function or replaced (using gene therapy or the direct administration of the normal chaperone). While these objectives may at the present time seem quite difficult to achieve, the reality is that progress in chaperonology has been, and continues to be, so fast that it is safe to predict that chaperonotherapy will be with us sooner than expected.

## Figures and Tables

**Figure 1 brainsci-08-00110-f001:**
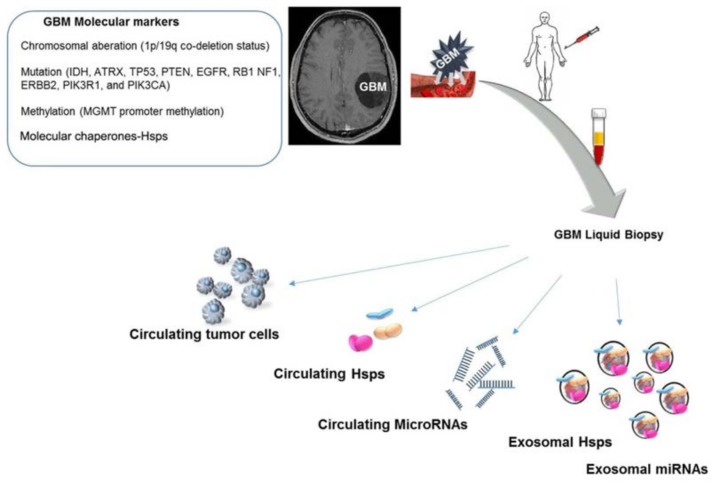
In the last few years, several genetic and molecular factors (some of which are listed in the top left inset) have been identified as pathogenic in glioblastomas multiforme (GBM). Through “liquid biopsy” it is possible to examine circulating tumor cells as well as tumor-cell products, such as cell-free proteins and nucleic acids, e.g., miRNAs, and extracellular vesicles, e.g., exosomes (shown in the lower half of the figure). Exosomes carrying heat shock proteins (Hsps) or Hsp-regulatory miRNAs have recently attracted interest, becoming novel biomarkers for diagnostics, and for assessing prognosis and response to treatment in various types of cancer, such as GBMs. Because of the minimal invasiveness of the procedure and its low cost, the quantification and characterization of Hsp and Hsp-carrying exosomes are very promising tools in clinics.

**Figure 2 brainsci-08-00110-f002:**
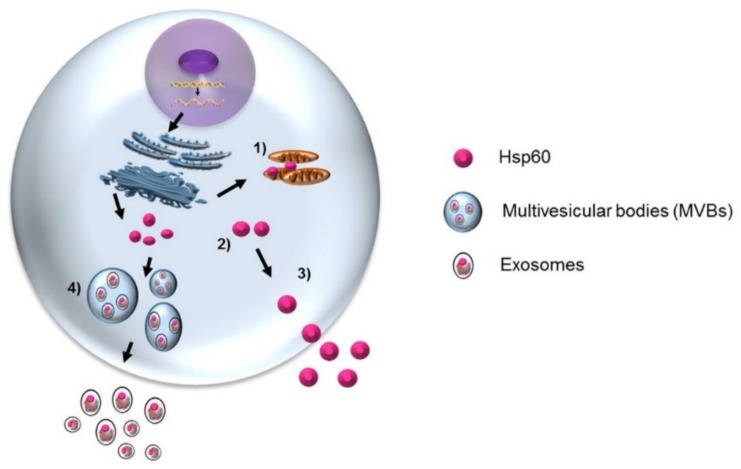
Hsp60 is classically a mitochondrial molecule (1) but it is found also outside the organelle, and various other places, such as in the cytosol (2), plasma-cell membrane, intercellular space, and blood. Its functions are therefore varied in physiology and pathophysiology, depending on where it resides. The levels of Hsp60 may be elevated or decreased in various types of cancer, and they are associated with tumor progression in some instances. It seems to have antitumor or protumor effects depending on the type of cancer and other conditions. Our research group has shown that tumor cells release Hsp60 via both the classical secretion pathway (3) and in multivesicular bodies-exosomes (4), and could thereby modulate the antitumor immune response, although this is still under investigation. The precise role of Hsp60 in brain tumor pathogenesis is still incompletely understood and more studies are necessary before all of the promising aspects of the chaperonin in what pertains to its value as a biomarker for diagnosis, assessing prognosis and response to treatment, and to its possible applications as therapeutic target or agent, can be fully exploited.

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
