# Peer review of "Chaperonology: The Third Eye on Brain Gliomas"

_brainsci, 2018, doi:10.3390/brainsci8060110_

Round 1

Reviewer 1 Report

Summary

This review article summarizes the pre-operative and intra-operative methodology in brain carcinogenesis and discusses the importance of molecular chaperon heat shock proteins (Hsps) as a potential biomarker in high grade gliomas.

Major comments

Authors should overall state why their review article is significant and how it is different than the other published reviews in the field. The authors should explain in detail the importance of heat shock protein in introduction section.

Authors have shown previously that Hsp60 levels is regulated in colon cancer, however no evidence is provided if  it is also differentially regulated in gliomas. Are there any previous studies showing that Hsp60 is differentially regulated in brain carcinogenesis?

A figure showing that Hsp60 is regulated in glioma is absolutely needed to conclude the study.

Minor Comments:

The manuscript needs improvement in grammar and syntax. It should be carefully edited to facilitate coherency.

Author Response

We thank the reviewer for this positive comment. We appreciate the suggestion and we have added some sentences according to the reviewer’s comment in the Introduction section (lines 61-70; 72-78, 79-82). Moreover, in relation to the question: “Are there any previous studies showing that Hsp60 is differentially regulated in brain carcinogenesis?” We have annotated two studies concerning the possible dysregulation in brain carcinogenesis (lines 235-237; 239-243).

Concerning the picture showing glioma regulation by HSP60, it was a great pleasure for us, to better depict this phenomenon through a figure representing this in detail (Figure 2)

Reviewer 2 Report

The authors summarize the current SOC for high-grade gliomas and also offer a literature review on the advancements in molecular profiling of gliomas. The review is very well written and provides a great overview of the field. The one thing I recommend that the authors consider adding is the limitation of molecular profiling of gliomas. While there has been a burst of new biomarkers for various cancer types, in most cases suitable drugs targeting those genes/ mutations are still in clinical trials or in early development.

Author Response

We are grateful with the reviewer for his/her comments and helpful suggestions. At the end of the manuscript a section “Limitations of the study” has been added, as requested.

Reviewer 3 Report

This excellent review gives a good overview on the intricate issue of the function of chaperones in high-grade gliomas. The introduction gives a good overview on high-grade glioma therapy. Furthermore, all relevant publications are being discussed. This work will find the interest of many readers. following issue should be adressed:

) The linguistic style should be improved by a thorough revision.

) In line 352, the IMET number is missing.

) The author contributions are missing.

Author Response

We really appreciate the reviewer’s comment. It’s a pleasure for us to find agreement in the scientific community. The linguistic style has been revised and corrected. The IMET number is missing since it can be added only after the manuscript acceptance.

A section with the contributions of each author has been added at the end of the manuscript.

Round 2

Reviewer 1 Report

Authors have addressed all the comments.